# Using Deep Learning to Detect Motor Impairment in Early Parkinson's Disease from Touchscreen Typing

Sophia Gu[*†]
Stony Brook University

Yan Ma[*‡]
Stony Brook University

Zhi Li[§]
Stony Brook University

Xiangmin Fan[¶]
Chinese Academy of Sciences

Feng Tian[‖]
Chinese Academy of Sciences

Xiaojun Bi[**]
Stony Brook University

## ABSTRACT

Assessing older adults' motor control ability is crucial for early diagnosis of Parkinson's Disease (PD). In this paper, we investigate how to use deep learning to detect motor impairment of older adults via analyzing touchscreen typing data, which could result in the early diagnosis of PD. Our investigation shows that deep learning is promising in analyzing touchscreen typing data. Among the four deep learning models (LSTM, LSTM-CNN, CNN-LSTM, and 1D CNN), LSTM-CNN yields the best performance. On a 102-subject dataset [53], LSTM-CNN achieved an AUC of 0.95 and an F1-score of 0.90 in leave-one-out PD classification, improving the performance of previously used SVM method (AUC = 0.88, F1-score = 0.73) [53]. LSTM-CNN also performed well on an in-the-clinic typing dataset [27] (AUC = 0.86, F1-score = 0.86), and significantly improved the F1-score of the previously proposed 1D CNN method (AUC = 0.89, F1 = 0.80). The promising performance of the LSTM-CNN model can also be generalized to other touchscreen interactions including flick, drag, handwriting, and pinch. It achieved better performance than the previous SVM method [53]. Our research showed that deep learning is effective in detecting early PD's motor symptoms via analyzing smartphone interaction data, and the proposed LSTM-CNN model is a promising neural network structure for performing such analysis. Overall, our research advances the understanding of how to assess the motor control ability of older adults via smartphone interactions.

**Index Terms:** Human-centered computing—Human computer interaction (HCI)—Interaction techniques—Text input; Computing methodologies—Machine learning—Machine learning approaches—Neural networks

## 1 INTRODUCTION

Parkinson's Disease (PD) is a chronic neurological disorder that causes progressive disability which can cause significant physical and mental impairment and decreased quality of life [52]. PD is now the most prevalent movement disorder in older adults: it affects more than one million Americans and more than 6.3 million people in the world [11, 32].

Early detection of motor impairments associated with PD is crucial for providing early intervention and treatment. However, as

---

[*]Both authors contributed equally to this research.

[†]sg.sophiag@gmail.com. Sophia Gu was a high school research intern supervised by Yan Ma and Xiaojun Bi at Stony Brook University.

[‡]yanma1@cs.stonybrook.edu

[§]zhili3@cs.stonybrook.edu

[¶]xiangmin@iscas.ac.cn

[‖]tianfeng@iscas.ac.cn

[**]xiaojun@cs.stonybrook.edu

frequent clinic visits could increase the physical and economic burden on patients and their families, patients may not seek professional assessment or clinical visits until the symptoms become severe. For example, a prior study [48] showed that most presentations of motor impairments associated with PD occurred within 2 years before the first clinical diagnosis of PD, and the incidence of the tremor was already higher in PD subjects compared with the control group at up to 10 years before diagnosis. Having a screening method that can detect motor symptoms of PD without clinic visits is of great interest to older adults and our society.

One promising direction for detecting motor symptoms of PD is to analyze typing data on a soft keyboard of a smartphone, because of the following reasons. First, typing requires intensive and consecutive target acquisition on a soft keyboard, which exposes the motor control ability of a user during a short period of time. Second, typing is one of the most common activities people perform daily on smartphones. Emailing, messaging, social networking, and searching are common tasks on smartphones that all involve intensive typing on soft keyboards. Thanks to its prevalence, it is possible to assess and monitor a user's motor control ability via analyzing his/her touchscreen typing data [26]. Indeed, previous research [53] has shown that using the traditional SVM machine learning methods to analyze the typing data could reach an F1-score of 0.73 and AUC of 0.88, in detecting the motor impairment of early PD users.

Although the previous work [53] showed the feasibility of detecting the motor impairment via touchscreen typing, the accuracy of the detection is unsatisfying: the AUC value was 0.88, and the sensitivity (true positive) value was only 0.7. One weakness of the previous work [53] was that it used SVM methods for classification, which might be suboptimal compared with the state-of-the-art deep learning models.

In this research, we aim to understand whether using deep learning models would improve the accuracy of detecting motor impairment of early PD users. Although it is expected that deep learning models would likely outperform the SVM methods, it is nontrivial to find out which model structure work bests for analyzing the touchscreen typing data, given various deep learning model structures proposed in the past decade. Furthermore, it is also important to understand whether the proposed deep learning model could be generalized to different typing datasets, and other types of touchscreen interaction. These are the research questions we aim to answer in this paper.

We investigated the effectiveness of four deep learning models in detecting PD motor symptoms, including Long Short-term Memory-Convolutional Neural Network (LSTM-CNN), CNN-LSTM, 1D CNN, and LSTM only. Our investigation on two existing datasets shows it is appropriate to use deep learning models to analyze typing data. In particular, the proposed LSTM-CNN model in which a recurrent block is followed by a convolutional block, substantially improved the classification performance over the SVM method adopted by the previous work [53] on the same 102-subject dataset: the LSTM-CNN model achieved an AUC of 0.95 and an F1-score of 0.90 in leave-one-out PD classification, improving the perfor-

mance of previously used SVM method (AUC = 0.88, F1-score = 0.73) [53]. This LSTM-CNN model also outperformed other three deep learning models including an LSTM model, a previously used 1D CNN model [24, 25] and a CNN-LSTM model [46] which is commonly used in video analysis [14, 31]. This LSTM-CNN model also performed well in analyzing an in-the-clinic typing dataset [27], which contained normalized pressure, timestamps, touch-down/up events, but no x-y coordinate data. It achieved an AUC of 0.86 and an F1-score of 0.86. Furthermore, the promising performance of the LSTM-CNN model can also be generalized to other touchscreen interaction data. It outperformed the previous SVM method in analyzing pattern drawing, handwriting, and sliding data. Overall, our investigation shows that deep learning is a promising method for detecting motor impairment of older adults in touchscreen typing, and the proposed LSTM-CNN model is a promising structure for performing such analysis.

## 2 RELATED WORK

### 2.1 PD Assessment

Since the treatment of PD requires regular and close monitoring related motor function symptoms, designing mobile tools for remote monitoring of PD symptoms has attracted a lot of attention. The eddy-current detector [13], portable multichannel recorder [23], wearable sensor [57], and inertial sensor [45] have been used to measure hand tremors.

With the rapid development of mobile computing technology and wide adoption of mobile devices, using smartphones to measure various movement-related metrics of PD patients has attracted a lot of research interest. For example, Fontecha et al. [17] used tri-axels accelerometers in smartphones to assess frailty in elderly people. Liddle et al. [36] used the global positioning system (GPS) sensor to evaluate lifespace of people with PD. Galan-Mercant et al. [18] utilized the accelerometer and gyroscope to measure sit-to-stand posture transition in elderly persons. Pan et al. [40] developed a mobile App to collect and analyze PD-related motion data using the smartphone 3D accelerometer. Lee et al. [33] created a smartphone APP to conduct a timed finger-tapping test (SmT) to monitor bradykinesia. These methods require patients either to wear extra devices or sensors which are obtrusive and introduce new cost [13,23,45,57], or operate the devices/software with specific procedures which requires learning [10, 20, 33, 39].

Our research is particularly related to previous work that detects motor impairment of older adults via analyzing touchscreen typing. Tian et al. have performed an comprehensive feature analysis and applied the SVM method to analyze the touchscreen typing data of 102 users (35 PD and 67 controls) [53]. Their method reached an AUC of 0.88 and an F1-score of 0.73. Iakovakis et al. extracted features from in-the-clinic typing data collected from 33 users (18 PD and 15 controls), and applied linear regression for classification [26]. Their method reached an AUC of 0.84 with an estimated rigidity index and 0.80 with an estimated brady-/hypokinesia index on an in-the-wild harvested dataset. Inspired by the recent progress in deep learning techniques, which eliminate the need for heavy ahead-of-time feature engineering, we investigated using deep learning models to analyze the typing data.

### 2.2 Pre-screening PD with a Deep Learning Method

In the recent past, many studies have been carried out to explore the effectiveness of deep neural networks on PD pre-screening. Previous research applied deep learning methods to different types of PD data, such as electroencephalogram (EEG) signals [34], underfoot sensor [4,56], wearable on body sensors [3,54], drawing [7], handwriting [2,19], speech signals [1,28,44], and smartphone sensor data and typing dynamics [24,25,41]. A few previous studies detected the presence of early PD symptoms via analyzing a combination of

several aforementioned types of data [35, 43, 55]. However, acquiring certain types of data (i.e., EEG signals, underfoot sensor, and wearable on body sensor data) requires devices that are not always accessible, and thus could not passively pre-screen PD symptoms in users' daily life. Since smartphones are ubiquitous nowadays, detecting motor impairment in early PD via analyzing smartphone typing dynamics becomes more and more promising. Papadopoulos et al. [41] proposed a deep learning framework that unobtrusively detected tremor, FMI, and PD from passively captured multimodal sensing data from smartphone. Iakovakis et al. [25] applied a one dimensional Convolutional Neural Network (CNN) with two input channels on both in-the-clinic and in-the-wild collected datasets.

Inspired by previous works, we further investigate whether different types of deep learning models could improve the performance, including the previously proposed 1D-CNN model [25] as a baseline. Our investigation shows the pros and cons of different models, and contributes a new LSTM-CNN model which has promising performance on different types of datasets, including the Tian et al's dataset [53] with touch coordinates, accelerometer and gyroscope data and Iakovakis' dataset [27] which has no touch-coordinates, accelerometer, or gyroscope data.

### 2.3 Time Series Classification

Researchers have proposed hundreds of algorithms to solve the Time Series Classification (TSC) problem [5]. Previous work has defined various new elastic distance measures for nearest neighbor classifiers to solve the TSC problem [29, 51]. Other work in this domain explored dictionary [37, 47] and interval-based approaches [12]. Researchers also proposed transformation-based techniques [22, 30] and ensemble methods [6, 38] to perform time series classification.

Recent work using deep neural networks achieved accurate results for TSC problems. In [59], a 1D CNN has been used for multivariate time series classification where filters are applied on each channel and features are flattened across channels as input to a fully connected layer. Cui et al. proposed a CNN-based approach for univariate time series classification [9]. Fawaz et al. proposed the InceptionTime [15], an ensemble of deep Convolutional Neural Network models, for TSC problems. In recent works [14, 31, 50, 58] CNN-LSTM based models were widely adopted to process, classify, and label video (time sequence of images) and text data (time sequence of words).

## 3 DEEP LEARNING MODELS

We considered four different deep neural networks: an LSTM-CNN model, a CNN-LSTM [46], a previously used 1D CNN based model [24, 25] with multiple input channels, and an LSTM model.

### 3.1 Problem Definition

We formulated the early PD detection problem as a binary classification problem: Given a sequence of touch events generated during the user typing on a smartphone touchscreen, predict whether the user is diagnosed with PD or not. For a user $i$, his/her touch events sequence is denoted by a vector $X_i = \{\mathbf{x}_1, \mathbf{x}_2, ..., \mathbf{x}_T\}$, $\mathbf{x}_t \in \mathbb{R}^d$, where $\mathbf{x}_t$ denotes the $d$-dimensional raw feature vector collected at time $t$. We also used a label $Y_i$ to indicate whether the user is diagnosed with PD, i.e. $Y_i = 1$ indicates the $i^{th}$ user is diagnosed with PD while $Y_i = 0$ means the user is free of PD. We observed $N$ random samples $(X_1, Y_1), (X_2, Y_2), ..., (X_N, Y_N)$ from $N$ distinct users in a dataset. Our goal is to predict the label $Y_i$ for any given input $X_i$ in the $N$ samples.

### 3.2 LSTM-CNN Model

We first proposed an LSTM-CNN neural network model containing the following layers (Fig. 1).

- Input Layer: It inputs the initial data to the neural network and sets its dimensions.

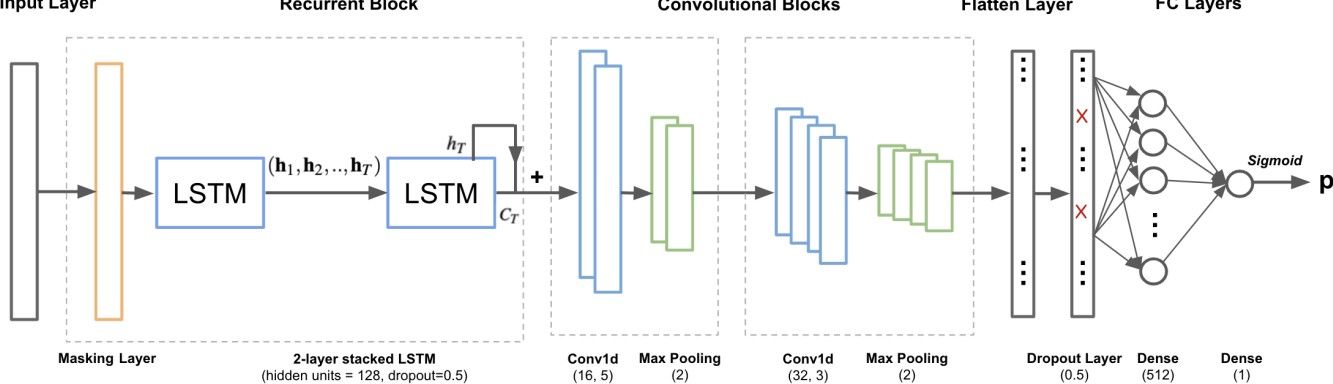

Figure 1: The proposed LSTM-CNN model contains an input layer, a recurrent block, two convolutional blocks, a flatten layer, a dropout layer, and two fully connected layers. It takes a sequence of $d$-dimensional raw feature vectors and outputs a probability score $p$ for the positive class. The LSTMs are firstly used to process both local and global time-related information. The CNN layers then extract features based on spatial relations in the LSTM output for classification.

- Recurrent Block: It consists of a masking layer and a two-layer stacked LSTM.

    The masking layer allows the model to identify and ignore all missing time-steps by using a mask value, and to skip these time-steps during processing the data. It enables the LSTM layers to accept input sequences with different lengths.

    The stacked LSTM contains two LSTM layers, each consists of $K$ hidden units. An LSTM layer takes a sequence (length $= T$) of $d$-dimensional feature vectors $(\mathbf{x}_1, \mathbf{x}_2, .., \mathbf{x}_T)$, $\mathbf{x}_t \in \mathbb{R}^d$ as input, and generates a higher-order feature representation at each time-steps, $(\mathbf{h}_1, \mathbf{h}_2, .., \mathbf{h}_T)$, where $\mathbf{h}_t \in \mathbb{R}^K$. In a two-layer stacked LSTM, the input of the second LSTM layer is the higher-order features generated by the first LSTM layer. We concatenated the last hidden state $h_T$ and the last cell state $C_T$ from the second LSTM layer to form the output of the reccurrent block, as shown in Fig. 1.

- Convolutional Block: It contains a convolutional layer and a max-pooling layer. The stacking of convolutional blocks allows a hierarchical decomposition of the features extracted by the recurrent block.

    The 1D convolutional layer applies filters to its input and creates a feature map that summarizes detected features. We used *tanh* as the activation function.

    The 1D max-pooling layer calculates the maximum value for each patch of the previous feature map and outputs a new feature map containing the most prominent features.

- Flatten Layer: It collapses the dimensions of the extracted features to a vector for classification.

- Dropout Layer: It prevents the model from overfitting by randomly setting input units to 0 at a rate at each step during training.

- Fully Connected Layer: It is a standard feed-forward layer connecting the convolutional block to the scaler output. The last fully connected layer compiles the features extracted by previous blocks and computes the binary cross-entropy loss to predict the final classification output. The total loss (*Loss*) is calculated as follow,

$$Loss = -\frac{1}{n} \sum_{i=1}^{n} (y_i \cdot \log p_i + (1 - y_i) \log(1 - p_i))$$

where $p_i$ is the scalar value of the $i^{th}$ model output, $y_i$ is the corresponding class label, and $n$ is the sample size.

We designed this LSTM-CNN model in order to leverage the advantages of both LSTM and CNN. LSTMs, as a special type of RNN, are able to capture temporal dynamics, and also enable arbitrary lags between important time events, which make them well-suited for processing and classifying time series data. CNN is capable of capturing distinctive features from input that has spatial relations. Stacked layers of CNN can extract discriminative feature maps in a hierarchical manner. Therefore, we expect the LSTM-CNN network structure can extract high-quality features that also contain time-related information, making the LSTM-CNN model suitable for analyzing and classifying typing data.

### 3.3 CNN-LSTM Model

As CNN-LSTM is also a common model for analyzing time-series data [46], we swapped the order of LSTM and CNN blocks in the LSTM-CNN model to form a CNN-LSTM model as another option.

We also made the following adjustments to make the CNN-LSTM model work. We configured the CNN-LSTM model to contain two time-distributed convolutional blocks, a recurrent block, a dropout layer, and two fully connected layers in sequence. In convolutional blocks, convolution is applied to raw feature vectors at each time step, and the output is passed through a time-distributed flatten layer before being processed by the recurrent block. The kernel sizes, pooling stride, dropout rate, and hidden units in CNN-LSTM network components are the same as the LSTM-CNN model in Section 3.2.

This CNN-LSTM model first exploits CNN to extract features based on spatial relations in the input, and reduces the dimensions of the sequence. Then, a stacked LSTM further extracts time-related dependencies in the data from the processed sequence. Finally, the fully connected layers compile the output of LSTM and compute the probability score for classification. This architecture is appropriate for time sequences that have a spatial structure in their data dimensions, such as the 2D structure of pixels in an image, or the 1D structure of words in a paragraph.

### 3.4 1D CNN Model

We also included the previously proposed 1D-CNN model [25] as the third option. The 1D CNN model contains two convolutional blocks, a flatten layer, a dropout layer, and two fully connected layers. We also used the same kernel sizes, pooling stride, and dropout rate as in the convolutional blocks of the LSTM-CNN model. The 1D CNN

| Event | Timestamp | Pressure | Touch point $x, y$ coordinates | Accelerometer values | Gyroscope values |
|---|---|---|---|---|---|
| Touch-up/down | $t$ | $p$ | $(x, y)$ | $(acc_1, acc_2, acc_3)$ | $(gy_1, gy_2, gy_3)$ |

Table 1: Raw data of each touch record. A touch record contains touch-down and touch-up event. 10 features were recorded at each touch event.

| Time | Distance | Speed | Acceleration | Other first derivatives |
|---|---|---|---|---|
| $HT, FT$ | $d, d_x, d_y$ | $v, v_x, v_y$ | $a, a_x, a_y$ | $\frac{dp}{dt}, \frac{dacc_1}{dt}, \frac{dacc_2}{dt}, \frac{dacc_3}{dt}, \frac{dgy_1}{dt}, \frac{dgy_2}{dt}, \frac{dgy_3}{dt}$ |

Table 2: Additional features added to the input vector of the neural network. $HT, FT$ are the hold time and flight time defined in previous work [26]. $d_x$, $d_y$ are distances in $x$, $y$ directions while distance is in the 2D-pixel coordinates. $v_x, v_y / a_x, a_y$ are speed/acceleration in $x, y$ directions. First derivatives of other features over time are also considered in the input feature vector.

model treats dimensions in a raw feature vector as channels of the input, and applies filters to perform 1-dimensional convolution along the temporal axis across all channels. It is capable of detecting local patterns in the temporal axis and extracting features based on those temporal relations.

### 3.5 LSTM Model

The LSTM model is the LSTM-CNN model in Section 3.2 without convolutional blocks and the flatten layer. It exploits LSTMs to extract important long and short-term features for classification in the final fully connected layer. We included it as an option to evaluate whether adding a CNN block after LSTM would improve the performance.

The four aforementioned deep learning models have components at the same scale, and thus can be compared in early PD classification. We implemented the four deep learning models using the Keras Python deep learning API [8].

## 4 RESULTS

We evaluated the performance of the four deep learning models on Tian et al.'s smartphone touch interactions dataset [53] and Iakovakis et al.'s in-the-clinic typing dataset [27], and compared the results with a baseline SVM model [53].

### 4.1 Evaluation on Tian et al.'s Typing Data [53]

Tian et al. collected the PD smartphone touch interactions dataset to investigate the feasibility and performance of detecting early PD motor impairment from smartphone typing and common gestures [53]. In total, they collected the touch events sequences of 102 subjects, 45 years or older, during a transcribing user study. Among the 102 subjects participating in this study, 35 were early PD patients, and 67 were age-matched healthy controls. All of the patients recruited in their study were at early PD stages (Hoehn-Yahr stages I or II[2], mean UPDRS Part III score/std 8.4/3.7) [21]. The dataset includes raw features such as touch point pixel coordinates, timestamps, pressure, and measurements from the accelerometer and the gyroscope. It investigated typing and four common gestures, including flick, drag, handwriting, and pinch. For typing tasks, excerpts were selected from a Chinese short-text conversation corpus [49]. We used the 2266 touch records collected from 101 subjects (35 PD patients and 66 controls) typing on a custom QWERTY keyboard. Each touch record contains a touch-down event and a touch-up event, and contains the following information as shown in Table 1. In total, we used data of the 4532 touch events in this dataset.

### 4.1.1 Input Features

The input to all four deep learning models is a sequence of 36-dimensional raw feature vectors describing each touch record. In addition to the 18 dimensions (9 for each touch event, timestamp and type not included) in Table 1, we also included features in Table 2 to form the input to the network.

| Model | AUC [5%, 95%] | Sens | Spec | F1 |
|---|---|---|---|---|
| SVM | 0.88 [0.80, 0.93] | 0.74 | 0.85 | 0.73 |
| **LSTM-CNN** | **0.95 [0.90, 1.00]** | **0.89** | **0.96** | **0.90** |
| 1D CNN | 0.92 [0.86, 0.99] | 0.86 | 0.85 | 0.80 |
| CNN-LSTM | 0.85 [0.75, 0.92] | 0.80 | 0.87 | 0.78 |
| LSTM | 0.94 [0.88, 1.00] | 0.86 | 0.96 | 0.88 |

Table 3: Classification results of the five models on Tian et al.'s typing data. The four NN models have better performance than the SVM method. In particular, the LSTM-CNN network outperformed other models in terms of AUC, sensitivity, specificity, and F1-score.

### 4.1.2 Classification Results

We performed leave-one-out cross-validation (LOOCV) on the four deep learning models and the SVM model [53]. More specifically, in each pass, we trained the four deep learning models with the data from 101 subjects and predicted the probability scores of the excluded one subject. In this manner, the results obtained below are subject-independent. Similar to previous work [53], we evaluated and compared the aforementioned models using the following metrics: sensitivity (i.e. true positive rate), specificity (i.e. true negative rate), F1-score (i.e. the harmonic mean of precision and recall), Receiver Operating Characteristic (ROC) analysis, and Area under the ROC Curve (AUC). These metrics provide a consistent measurement even given imbalanced class sizes. More specifically, we computed the ROC curve by plotting sensitivity against $(1 - \text{specificity})$ (i.e. false positive rate) at different classification thresholds [16, 24]. We estimated the distribution of the ROC curve and calculated the average and 95% confidence interval of the AUC values with 1000 bootstraps. In addition, we applied the closest-to-$(0, 1)$ criterion to define the cut-off point (i.e. decision threshold) for calculating sensitivity, specificity, and the F1-score [42].

Table 3 summarizes the results of SVM with manually extracted features [53] and the four deep neural network models. The LSTM-CNN model achieved an AUC of 0.95 [0.90, 1.00] and an F1-score of 0.90 in discriminating early PD. Both values outperform other models, indicating that LSTM-CNN is more suitable for pre-screening early PD with smartphone touchscreen typing data. A higher sensitivity over other models enables the LSTM-CNN network to better identify PD motor impairment. The LSTM-CNN model also has the highest specificity of 0.96 among all five models, meaning that false alarms of PD are rare compared to other methods. A pre-screening test with low specificity may not be applicable, since many users without PD will screen positive, and potentially receive unnecessary diagnostic procedures that add to their financial burden. Given that the LSTM-CNN model also achieved a decent sensitivity value, a high specificity of 0.96 is preferred in practice. Fig. 2 shows the ROC curve distributions estimated from different methods, and further demonstrates the classification performance of each model. The

mean ROC curve of the LSTM-CNN model is the closest to the $(0, 1)$ point among all five model candidates, indicating that LSTM-CNN outperformed other models in classifying PD.

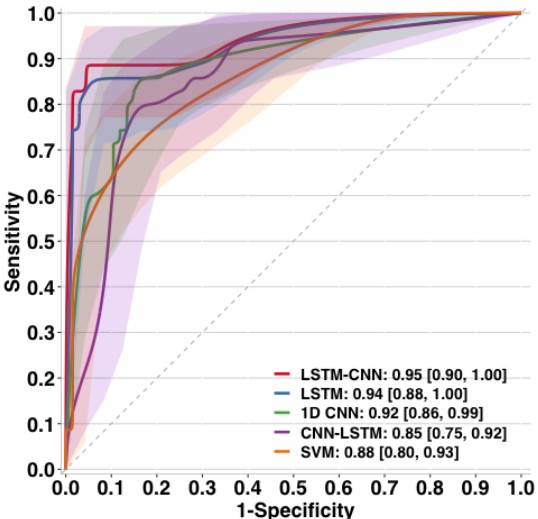

Figure 2: Smoothed ROC curves demonstrating the classification performance of the five models on Tian et al.'s typing data [53]. The curves show the mean ROC. The bands are the mean ROC $\pm$ 1 standard deviation. The grey dashed line indicates a random prediction. Mean ROC curves of the four deep learning models are closer to the $(0, 1)$ point compared to the SVM baseline, meaning that deep learning models have better classification performance. Among five ROC curves, the red one is closest to the upper left corner in the figure, indicating that LSTM-CNN outperformed other models in ROC analysis [42].

### 4.2 Evaluation on Iakovakis et al.'s In-the-Clinic Dataset

We further evaluated models on the Iakovakis et al.'s In-the-Clinic touchscreen typing Dataset [27] which has no touch point $(x, y)$ coordinates and smartphone sensor data. The in-the-clinic PD smartphone typing dataset was acquired from 33 subjects (18 PD patients and 15 healthy controls). It consists of smartphone typing data and a clinical evaluation for each subject. For the typing data, Iakovakis et al. collected the normalized pressure (0.000-1.000), as well as the timestamps at touch-down and touch-up events for each touch record on a QWERTY soft keyboard. However, touch point $(x, y)$ coordinates and smartphone sensor data were not included. We used data of the 26288 touch events from this dataset.

#### 4.2.1 Input Features

The input to all models is a sequence of 4-dimensional raw feature vectors, which contains pressure $(p)$, hold time $(HT)$, flight time $(FT)$, and the first derivative of pressure over time $(\frac{dp}{dt})$. $HT$ is the time difference between a touch-down event and its corresponding touch-up event, and $FT$ is the time difference between a touch-down event and its previous touch-up event [26].

#### 4.2.2 Classification Results

We followed the same procedures as in 4.1.2 to perform LOOCV on Iakovakis et al.'s in-the-clinic dataset. In each pass, we trained the four models with 32 subjects' data and predicted the probability scores for the excluded one. We calculated pressure-based features described in [53] and the min/max/mean/median/standard deviation values of $HT/FT$ as input to the linear-kernel SVM. For the 1D

| Model | AUC [5%, 95%] | Sens | Spec | F1 |
|---|---|---|---|---|
| SVM | 0.87 [0.75, 0.99] | 0.83 | 0.80 | 0.83 |
| LSTM-CNN | 0.86 [0.73, 0.99] | **0.83** | **0.87** | **0.86** |
| 1D CNN | **0.89 [0.78, 1.00]** | 0.79 | 0.79 | 0.80 |
| CNN-LSTM | 0.46 [0.26, 0.66] | 0.72 | 0.33 | 0.63 |
| LSTM | 0.73 [0.55, 0.90] | 0.78 | 0.73 | 0.78 |

Table 4: Classification results of the five models on Iakovakis et al.'s in-the-clinic typing dataset [26]. The 1D CNN model outperformed other models in AUC. The LSTM-CNN model performed best in terms of sensitivity, specificity, and F1-score.

CNN network, We followed Iakovakis et al.'s work [25] and generated similar results using a 1D CNN model with multiple input channels. Since the in-the-clinic dataset only contains 33 subjects, we reduced the complexity of the four neural networks to avoid overfitting with the following modifications: (1) we decreased the hidden size in LSTMs to 32; (2) we only used one LSTM layer; (3) we removed the first fully connected layer.

Table 4 shows the classification results on Iakovakis et al.'s in-the-clinic dataset. The LSTM-CNN model achieved the highest sensitivity, specificity, and F1 score at 0.83/0.87/0.86 respectively. In terms of AUC estimations, the LSTM-CNN model achieved an AUC of 0.86 [0.72, 0.99], which is lower than the results of the SVM model (0.87 [0.75, 0.99]) and the 1D CNN network (0.89 [0.78, 1.00]). The LSTM-CNN model shows advantages in sensitivity, specificity, and F1 score, but not in AUC. One possible reason is that Iakovakis et al.'s in-the-clinic dataset was small with only 33 subjects, which could not fully demonstrate the power of different deep learning models.

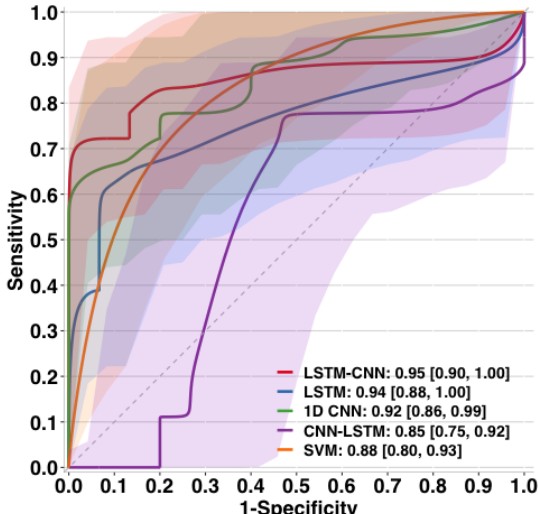

Figure 3: Smoothed ROC curves demonstrating the classification performance of the five models on Iakovakis et al.'s in-the-clinic typing dataset [53]. The curve shows the mean ROC. The band is the mean ROC $\pm$ 1 SD. The grey dashed line indicates a random prediction. The ROC curves of the LSTM-CNN and the 1D CNN model are closer to the upper left corner than the other three models, indicating a better classification performance [42].

Fig. 3 plots the ROC curve distributions estimated from different models. In terms of average ROC curves, the LSTM-CNN model generated a curve that is the closest to $(0, 1)$, the left top corner in the figure. A closer ROC curve means that the LSTM-CNN model outperforms other models in distinguishing PD typing data from

| Time | Pressure | Coordinates | Accelerometer | Gyroscope | Distance | Speed | Acceleration | Other first derivatives | | | | | | |
|---|---|---|---|---|---|---|---|---|---|---|---|---|---|---|
| $dt$ | $p$ | $(x,y)$ | $(acc_1, acc_2, acc_3)$ | $(gy_1, gy_2, gy_3)$ | $d, d_x, d_y$ | $v_x, v_y, v$ | $a, a_x, a_y$ | $\frac{dp}{dt}$, | $\frac{dacc_1}{dt}$, | $\frac{dacc_2}{dt}$, | $\frac{dacc_3}{dt}$, | $\frac{dgy_1}{dt}$, | $\frac{dgy_2}{dt}$, | $\frac{dgy_3}{dt}$ |

Table 5: Raw input features of single-handed gestures, including flick, drag, and handwriting. $dt$ is the time difference between timestamps of two adjacent touch records.

| Finger distance | Change of angle | Distance in x | Distance in y | Angular velocity | Finger distance velocity |
|---|---|---|---|---|---|
| $F_{dist} = \left\| \sqrt{x_1^2 + y_1^2} - \sqrt{x_2^2 + y_2^2} \right\|$ | $d\theta$ | $\left\| x_1 - x_2 \right\|$ | $\left\| y_1 - t_2 \right\|$ | $\frac{d\theta}{dt}$ | $\frac{dF_{dist}}{dt}$ |

Table 6: Additional features for the pinch gesture. Finger distance is the distance on the screen between touch points generated by two hands at the same timestamp. Change of angle is the change in rotation angle from the previous timestamp in pinch gestures.

healthy controls given the optimal decision threshold defined by Perkins and Schisterman [42] in practice. The results on Iakovakis et al.'s in-the-clinic dataset show that our LSTM-CNN model is promising in PD pre-screening via smartphone typing data even with limited information.

### 4.3 Evaluation on Tian et al's Touch Gestures Data [53]

#### 4.3.1 Touch Gestures

Although this paper mainly focuses on analyzing touchscreen typing data, we further analyzed the model performance on other touch-screen interaction data, to understand whether the benefits of deep learning models could be generalized beyond touchscreen typing. We evaluated the four deep learning models on touch gestures in Tian et al's dataset and compared its performance with the SVM model [53]. Tian et al. collected four common types of gestures that approximately cover daily touch gestural interactions on smartphone touchscreens, including flick, drag, handwriting, and pinch.

Since the four common gestures are also composed of touch input sequences, detecting PD motor impairment via analyzing common smartphone gestural interactions can also be defined as a binary classification problem on time series. Therefore, aforementioned four deep learning models for typing can be generalized to common smartphone gestures with modifications in input dimensions. For touch gestures that are performed with one hand, i.e. flick, drag, and handwriting, we define a 26-dimensional raw feature vector in Table 5 for every touch event regardless of its type (touch up/down/move). For bimanual gestures like pinch, we added pressure, the first derivative of pressure, touch point $(x,y)$ coordinates, distance, speed, and acceleration on top of Table 5 for the second hand (13 dimensions). We also considered another 7 features in Table 6. In total, the input vector of the pinch gesture has 46 dimensions. (Fig. 4) [53].

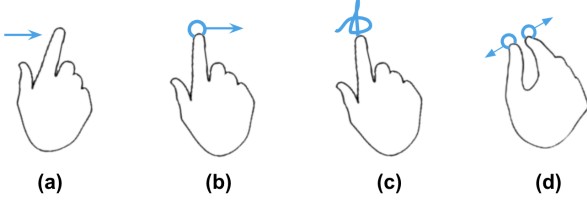

Figure 4: Common gestures defined in [53]. (a) flick gestures; (b) drag gestures; (c) handwriting gestures; (d) pinch gestures.

#### 4.3.2 Classification Results

We performed LOOCV on the data of the four gestures individually, so that the results obtained below are user-independent. Table 7 summarizes the performance of the five models on the four common

gestures. The LSTM-CNN also showed promising performance. It outperformed the other four models for flick and handwriting in AUC values and performed the best among all five models in terms of F1-score for all types of gestures. The LSTM-CNN model also achieved the highest specificity in every gesture, indicating that our model will generate fewer false alarms in early PD pre-screening tests. The sensitivity value of our model is also the best for flick and handwriting, making it competent in effectively detecting true positive cases.

| Gesture | Model | AUC [5%, 95%] | Sens | Spec | F1 |
|---|---|---|---|---|---|
| | SVM | 0.88 [0.80, 0.94] | 0.77 | 0.75 | 0.68 |
| | **LSTM-CNN** | **0.91 [0.85, 0.98]** | **0.89** | **0.88** | **0.84** |
| Flick | LSTM | 0.89 [0.82, 0.97] | 0.86 | 0.88 | 0.82 |
| | CNN-LSTM | 0.51 [0.34, 0.62] | 0.60 | 0.49 | 0.47 |
| | 1D CNN | 0.89 [0.82, 0.97] | 0.86 | 0.82 | 0.78 |
| | SVM | 0.92 [0.84, 0.96] | **0.86** | 0.88 | 0.82 |
| | LSTM-CNN | 0.92 [0.84, 0.96] | 0.83 | **0.93** | **0.84** |
| Drag | LSTM | **0.92 [0.85, 0.98]** | 0.83 | 0.90 | 0.82 |
| | CNN-LSTM | 0.83 [0.74, 0.92] | 0.80 | 0.81 | 0.74 |
| | 1D CNN | 0.84 [0.75, 0.93] | 0.80 | 0.81 | 0.74 |
| | SVM | 0.89 [0.82, 0.94] | 0.71 | 0.85 | 0.71 |
| | **LSTM-CNN** | **0.90 [0.83, 0.97]** | **0.89** | **0.88** | **0.84** |
| Handwriting | LSTM | 0.90 [0.83, 0.97] | 0.83 | 0.84 | 0.77 |
| | CNN-LSTM | 0.73 [0.63, 0.84] | 0.83 | 0.57 | 0.62 |
| | 1D CNN | 0.88 [0.80, 0.96] | 0.83 | 0.82 | 0.76 |
| | SVM | **0.92 [0.85, 0.96]** | 0.80 | 0.82 | 0.75 |
| | LSTM-CNN | 0.86 [0.77, 0.94] | 0.80 | **0.84** | **0.76** |
| Pinch | LSTM | 0.85 [0.77, 0.94] | 0.80 | 0.79 | 0.73 |
| | CNN-LSTM | 0.81 [0.72, 0.91] | **0.83** | 0.69 | 0.68 |
| | 1D CNN | 0.84 [0.75, 0.93] | 0.80 | 0.78 | 0.72 |

Table 7: Classification results of the five models on four common gestures on Tian et al's touch gestural interactions dataset [26]. The LSTM-CNN model outperformed other models on flick and handwriting gestures in terms of AUC and achieved the best F1-score among all five models.

In summary, this evaluation shows that deep learning methods can be easily generalized to other data, and can produce promising results which outperform the traditional SVM model. In particular, the proposed LSTM-CNN model is also a promising structure for using deep learning to detect early PD motor impairment with data from other touchscreen interactions in addition to typing.

## 5 DISCUSSION

### 5.1 Implicit PD Detection

Our results demonstrate that the deep learning models, especially the LSTM-CNN model performed well compared to the previously used SVM model [53] in distinguishing PD typing data from healthy controls. The LSTM-CNN model achieved an AUC of 0.95 and an F1-score of 0.90 for Tian et al.'s dataset [53], improving the performance achieved by the SVM approach (AUC = 0.88, F1-score = 0.73). The LSTM-CNN model also performed well on Iakovakis et al.'s in-the-clinic dataset [27], especially in terms of F1-score (AUC = 0.86, F1-score = 0.86). Moreover, the proposed LSTM-CNN model can be easily generalized to other smartphone interactions and achieved the best results among the aforementioned four models in AUC and F1-score for flick, drag, and handwriting gestures. These results indicate that deep learning is promising in detecting PD from touchscreen interaction data, and the proposed LSTM-CNN model advances the understanding of how to implicitly assess early PD motor impairment of older adults via smartphone interactions.

### 5.2 Deep Learning for Early PD Detection

Deep learning has several benefits over traditional machine learning methods, such as an SVM model or a logistic regression model. These benefits contribute to better performance on the PD classification problem. Firstly, a deep neural network fully utilizes the input data and delivers high-quality results. Compared to traditional methods such as SVM, deep neural networks also take into account the dependencies among different dimensions of the data, which leads to better empirical results because dimensions in real-world data are usually not independent. Secondly, neural networks also process data non-linearly. In practice, detection of early PD motor impairment via keyboard typing is probably not a linear problem. Therefore, its decision boundaries cannot be well approximated with linear representations in methods such as a logistic regression model. Moreover, deep learning neural networks eliminate the need for ahead-of-time feature engineering where input features are carefully determined through various statistical tests, such as features input to an SVM model. Descriptive features can be automatically extracted directly from the input data based on the desired outcome.

In addition, deep learning also has the following merits that make it a promising approach for detecting early PD motor impairment in future applications and research. Deep architectures are generalizable to fit different data and applications, and they are also flexible to be adapted for solving new problems. The proposed LSTM-CNN model can be applied to other PD typing datasets and adapted to classify PD on other collected time sequences rather than smartphone typing data. Since deep networks automatically learn variations in the input data, it also has fault tolerance to corruption in internal units and thus can generate robust results.

### 5.3 Performance and Limitations of LSTM-CNN Model

Our evaluation indicates that the proposed LSTM-CNN model is promising for detecting early PD motor impairment with smartphone interactions data. One possible reason is that the recurrent block in LSTM-CNN captures both local and long-term time-related information, and stacked layers of CNN further extract distinctive features from the LSTM output. In such a manner, the LSTM-CNN model automatically selects distinctive features that also contain temporal information, making it suitable for analyzing touchscreen typing data, which contains sequences of touch inputs.

However, the benefits of LSTM-CNN on a small dataset are not as strong as on a big dataset. For example, the LSTM-CNN model improves specificity, sensitivity, and the F1-score, but not the AUC score on Iakovakis et al's in-the-clinic dataset [27] which contains only 33 subjects and has no touch-coordinates, accelerometer, or gyroscope data. Another limitation of the LSTM-CNN model is that

we assume that possible patients are capable of typing on a smartphone. How learning disabilities due to genetic or neurobiological factors other than PD affect PD pre-screening using deep learning models remains unclear and is worth conducting future research.

## 6 CONCLUSION

In this paper, we investigated the effectiveness of using deep learning for detecting motor impairment in early PD by analyzing smartphone touchscreen typing data of older adults. We evaluated four deep learning models (LSTM-CNN, CNN-LSTM [46], 1D CNN [25], and LSTM only) on two datasets: Tian et al.'s dataset [53] and Iakovakis et al.'s in-the-clinic dataset [27]. Our evaluation shows that deep learning is an appropriate approach to analyze touchscreen typing data. In particular, the LSTM-CNN model is a promising model. It achieved an AUC of 0.95 and an F1-score of 0.90 in leave-one-out PD classification and outperformed the previous SVM method [53]. The LSTM-CNN model also performed well on Iakovakis et al.'s in-the-clinic dataset (AUC = 0.86, F1-score = 0.86) and significantly improved the F1-score of the previously used 1D CNN method. Furthermore, the promising performance of the LSTM-CNN model can be generalized to other types of touchscreen interaction data, including flick, drag, handwriting, and pinch. Overall, our research shows that deep learning is promising in detecting PD from touchscreen interactionss, and the proposed LSTM-CNN model is a promising neural network structure to perform such analysis.

### ACKNOWLEDGMENTS

We thank anonymous reviewers for their insightful comments and Iakovakis et al. for providing their dataset. This work was supported by NSF awards 2113485, 1815514, and NIH award R01EY030085.

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
