# OpenReview forum: "Using Deep Learning to Detect Motor Impairment in Early Parkinson’s Disease from Touchscreen Typing"
_graphicsinterface.org/Graphics_Interface/2022/Conference — GI 2022_

### Official Review · Reviewer_Z5WY · 2022-04-12
**Seems promising but hard to assess given the nature of the work**

**Rating:** 8
**Confidence:** 1

**Review:**

This paper compares four deep learning models for their use in early detection of Parkinson's disease (PD) from typing/computer input data. An existing dataset is used to train and compare the models.

HCI researchers have shown interest in recent years in understanding how computer input data can be used to detect early signs of diseases including PD. The contributions of this research stand to advance those efforts by demonstrating a superior approach to analyzing the data using deep learning. The paper is well written and easy to read despite the highly technical nature of the content. Overall, the research appears to have been robustly conducted and sound. Each of the methods is well explained as is the approach to evaluation and the results. As a reader, I find this paper well-targeted to an HCI audience interested in the application of AI to HCI problems. However, as reviewer entrusted with evaluating the correctness of the work, I lack confidence in my ability to evaluate whether the work uses appropriate methods and applies them correctly.

Accordingly I am a bit torn about my assessment of this paper. On the one hand, I think the HCI community is well placed to take up the results of this research and thus I'd like to see it published here (rather than in an AI conference where it might remain unnoticed by HCI researchers designing disease detection tools). On the other hand, I'm not certain this community will have a sufficiently robust expertise to properly vet the correctness of the work (I certainly do not). I appreciate that papers like this can be very difficult to home and that this is a challenge for authors of such papers. I also appreciate that there are risks to both to being too conservative in what we accept as well as too liberal. As such, I've rated the paper fairly high but my confidence low. My hope is that other reviewers will have the necessary expertise to confirm the correctness of the work (and round out my weaknesses), in which case I would support acceptance. However, if the reviewer confidence is across the board weak on the deep learning expertise necessary to assess the correctness of the work, then I would hesitate to recommend acceptance of this work and would suggest it should be submitted to a conference with a clearer expertise in the methods used in the paper.

---

### Official Review · Reviewer_zXQf · 2022-04-13
**The paper discusses a very important topic and how deep learning can be used to advance the domain of predicting early PD symptoms. However, I find the paper lacking major contribution as authors modified an existing model and evaluated it on existing datasets. I acknowledge that the findings might be promising to improve PD detection however further research needs to be conducted to understand the performance of the proposed model.**

**Rating:** 4
**Confidence:** 3

**Review:**

Summary:
Authors propose a deep learning model to conduct PD diagnosis by analyzing smartphone interactions. Results showes that the proposed model outperforms existing solutions to demonstrate a promising early detection of PD symptoms using neural network analysis.

Reasons to accept:
- The models adopted were tested on various datasets.
- The paper is generally well written and the literature review section summarizes existing relevant work.

Reasons to reject:
- I find the contributions limited since the proposed work relies on modifying an existing model and comparing it against existing solutions and evaluating the proposed model on existing datasets.
- The motivation behind introducing the LSTM-CNN model is not clear. It seems like a random change for the existing CNN-LSTM model right now. Authors need to provide more explanation of the motivation behind the introduced changes to the existing model.
- Authors claim that LSTM-CNN outperforms the rest of the models. However, this was not the case in the second dataset as well as for some gestures for the third dataset. I find the results showing a potential for using the LSTM-CNN for predicting PD symptoms however other experiments needs to be conducted in order to confim whether it outperforms the existing neural network models.

---

### Official Review · Reviewer_tegT · 2022-04-13
**Authors must address methodological kinks, content, context and language to make your paper easier to read, understand, replicate and be built on**

**Rating:** 6
**Confidence:** 3

**Review:**

The paper addresses a significant subject and uses technology used by a wide array of persons daily, making the advancement essential to help diagnose persons at risk of Parkinson's disease early.

A significant weakness in this article is the number of abbreviations used that are not given their full worded names, _LSTM, LSTM-CNN, CNN-LSTM, and 1D CNN_, for example. These make the paper jargon-filled and illegible for scientists of intersecting fields, which reduces the impact of your work to a smaller audience than it can truly reach. It lacks substance when connecting deep neural networks to the health subject, field and potential impact if done correctly. The paper also fails to address what happens if a person has learning disabilities or any other issues with typing that are not connected to PD, how do the models respond? What are the model's clear limitations?

The **introduction** makes a weak case of this research’s potential impact on the health of millions of people, patients, caregivers and medical practitioners who can benefit from it. Sell it without fear! Clean up the abbreviation saturation and put some effort to drop jargon whenever possible in this step of your research. Make clear that you are using previous data to evaluate/train your models.

The **related work** is inspired by previous research, which is excellent but doesn’t tell us what you are doing to go above and beyond this previous research with clarity. However, your research talks about and addresses people with an illness that can be devastating to get diagnosed with; please, do not refer to them as _users_. Potential patients, persons, and people are better words to refer to.

Please explain why the use of CNN sees a better result and promise when dealing with time-series data.

The **deep learning models (3)** can be defined, and their differences can be explained here, so we know why these four models and how they are different from one another. Lastly, we also know what you intended to achieve/compare with these other options.

When you talked about your LSTM-CNN Model, you say you have used the _tahn_ formula for the activation function, but you don’t tell us why. You did a great job explaining your model’s recipe. Great for scientific replication, fantastic!
It was unclear if your evaluation was _a training _of each model using Tian’s and Iakovaki’s data; please make it clear in your text what_ job_ each of these data sets performed.

In the **results sections 4.1.2 and 4.2.2**, you mention using previous work as a base for your classification and ROC curve. However,  you do not clearly explain these steps. Therefore, if the reader is not familiar with or cannot recall the cited work, the steps used to obtain the results are unknown.

Section 4.3 is a nice to have but not clear regarding:
1. Its connection and reliability when it comes to detecting PD
2. The procedure to attain classification results
3. Its reliability is considered necessary for generalising data in the field.
So, if possible, address the points above to make your argument tighter.

In your **discussion*, you mention that the  LSTM-CNN Model is clearly outperforming the others, which is valid for your first experiment but untrue for the following experiments. This is a major red flag which makes me wonder about your understanding of the different models and their performance. The affirmation that the LSTM-CNN Model was the best performing one overall is simply untrue.

In your **conclusion**, expand on the possibilities of future work beyond stating it is promising; if you cannot find any tangible examples of this bright future, it just reads like an empty statement and a weak end to such an exciting piece of technology you have built.

---

### Decision · Program_Chairs · 2022-04-17

Accept